# Genome-Wide Analysis and Expression Profiles of Auxin Response Factors in Ginger (*Zingiber officinale* Roscoe)

**DOI:** 10.3390/ijms26178412

**Published:** 2025-08-29

**Authors:** Yuanyuan Tong, Sujuan Xu, Jiayu Shi, Yi He, Hong-Lei Li, Tian Yu, Sinian Zhang, Hai-Tao Xing

**Affiliations:** 1College of Smart Agriculture/Institute of Special Plants, Chongqing University of Arts and Sciences, Chongqing 402168, China; 202306015003@stu.cqwu.edu.cn (Y.T.);; 2Chongqing Key Laboratory for Germplasm Innovation of Special Aromatic Spice Plants, Chongqing University of Arts and Sciences, Chongqing 402168, China

**Keywords:** ginger, *ZoARF*, growth, expression pattern, abiotic stress

## Abstract

Ginger, an economically important crop, fulfills multifunctional roles as a spice, vegetable, and raw material for medicinal and chemical products. The family of Auxin Response Factors (ARFs) plays an essential role in facilitating auxin signal transduction and regulating plant growth and development. However, the role of *ARF* genes in ginger, a crop of considerable economic importance, remains elucidated. In this study, a total of 26 *ZoARF* genes were identified in the ginger genome, which were further categorized into four subfamilies (I–IV) and displayed a non-uniform distribution across 11 chromosomes. The proteins are predominantly localized to the nucleus. Promoter regions contained numerous *cis*-elements linked to light signaling, phytohormones, growth, development, and stress responses. Collinearity analysis revealed 9 pairs of fragment duplication events in *ZoARFs*, all uniformly distributed across their related chromosomes. In addition, the expression profiles of *ZoARFs* in ginger were analyzed during development and under several stress conditions like ABA, cold, drought, heat, and salt, employing RNA-seq data and qRT-PCR analysis. Notably, expression profiling revealed tissue-specific functions, with *ZoARF#04/05/12/22* associated with flower development and *ZoARF#06/13/14/23* implicated in root growth. This work provides an in-depth insight into the *ARF* family and establishes a foundation for future investigations of *ZoARF* gene functions in ginger growth, development, and abiotic stress tolerance.

## 1. Introduction

Auxin, the first identified phytohormone, broadly regulates plant growth and developmental processes, including embryogenesis, vascular differentiation, lateral root initiation, floral morphogenesis, and apical dominance [1,2,3]. Auxin orchestrates plant development predominantly by regulating transcription in specific gene families, which include the *Auxin/Indole-3-acetic acid* (*Aux/IAA*), *Gretchen hagen 3* (*GH3*), *Small auxin-up RNA* (*SAUR*), and *Auxin response factor* (*ARF*) families [4]. Moreover, subsequent studies have demonstrated that these early auxin-responsive genes contain conserved promoter elements, including the TGA element (AACGAC), the core element of the auxin response region (AuxRE-core; GGTCCAT), and the auxin response element (AuxRE; TGTCTC) [5,6]. As a crucial component of the auxin signaling pathway, ARF proteins play a vital role in either activating or inhibiting auxin response genes by binding to AuxRE in the promoters [7]. A typical ARF protein features three conserved domains [8]. The first is the B3-type DNA-binding domain (DBD), positioned at the N-terminus, which recognizes AuxREs. The second region, termed the non-conserved middle region (MR), is located in the central portion of the protein and governs the activation or repression of auxin-responsive gene expression. This region is divided into an activation domain (AD) and a repression domain (RD). The AD is characterized by an abundance of leucine (L), glutamine (Q), and serine (S) residues, whereas the RD is rich in glycine (G), leucine (L), serine (S), and proline (P) residues [9,10]. The C-terminal domain (CTD) facilitates protein-protein interactions, enabling the formation of homodimers or heterodimers between ARF-AUX/IAA or ARF-ARF, thereby regulating auxin-responsive gene expression [11]. ARF-mediated transcriptional activity can be partially restricted by ARF transcription factors and Aux/IAA protein dimers when auxin levels are low. Conversely, elevated auxin concentrations promote the 26S proteasome to release ARF proteins from inhibitory dimer complexes, resulting in the induction of downstream auxin-responsive genes [12].

Genome-wide analyses across various plant species have provided a comprehensive characterization of the *ARF* gene family, revealing notable interspecies variation in gene family members: *Arabidopsis thaliana* (23) [13], *Oryza sativa* (25) [14], *Zea mays* (31) [15], *Citrus sinensis* (16) [16], *Prunus persica* (17) [17], *Musa acuminata* (47) [18], and *Panax ginseng* (53) [19]. Despite this broad phylogenetic distribution, biological function characterization of *ARF* remains predominantly concentrated on model species, particularly *Arabidopsis* and rice. So far, 22 *ARF* genes and one pseudogene have been characterized in *Arabidopsis* [10]. *AtARF* gene expression is dynamically and tightly regulated across various developmental stages and tissues. For instance, *AtARF1* is a transcriptional repressor. *AtARF2* regulates senescence and abscission of floral organs and leaves [20]. *AtARF1/AtARF2* double mutants exhibit phenotypes similar to single mutants but with enhanced traits, indicating partial functional redundancy of *AtARF1* and *AtARF2* [5,20]. *AtARF3/AtARF4* mediate development of both reproductive and vegetative tissues [21]. Additionally, *AtARF5* (MONOPTEROS) plays a crucial role in vascular tissue development [22]. Functional redundancy exists between *AtARF6* and *AtARF8*, which regulate reproductive capacity in pistils and stamens. *AtARF6/AtARF8* double mutants exhibit shortened stamen filaments and delayed anther dehiscence, resulting in female sterility [23]. In *AtARF7/AtARF19* double mutants, adventitious root formation is impaired, and lateral root number is reduced. However, neither of these single mutants significantly impairs adventitious or lateral root development, indicating functional redundancy between these genes [24]. In rice, *OsARF1* is known to regulate plant growth rate, leaf size, and plant dwarfism. It is also critical for nutritional and reproductive development [25]. Moreover, the expression changes in *OsARF11* and *OsARF15* under salt stress imply their participation in the rice stress response [26]. Collectively, *ARF* genes play a crucial role in different facets of plant development and stress responses in both *Arabidopsis* and rice, underscoring their significance in auxin signaling.

Ginger (*Zingiber officinale* Roscoe), a perennial herbaceous species in the Zingiberaceae family, has emerged as a globally significant cash crop predominantly propagated vegetatively through rhizomes [27]. These rhizomes accumulate bioactive compounds such as gingerols and shogaols, which confer characteristic pungency and diverse medicinal properties, including anti-inflammatory, analgesic, and immunomodulatory effects [28]. Ginger is extensively utilized in medicine and food processing due to its beneficial effects, such as improved digestion, nausea relief, anti-inflammatory and analgesic actions, and enhanced immunity [29,30].

Despite its dual significance as a medicinal and culinary crop, ginger faces persistent productivity constraints due to abiotic stressors, including high temperatures, drought, and soil salinity that collectively impair rhizome development and diminish yields [31,32]. Although *ARFs* have been widely characterized as key transcriptional regulators governing multiple developmental pathways in plants, their systematic identification and functional annotation remain unexplored in *Z. officinale*. Despite the recent establishment of a high-quality ginger genome assembly [33], enabling genomic exploration of *ARF* homologs, no comprehensive genome-wide analysis of the *ARF* gene family has yet been conducted in this species. To address this knowledge gap, this study conducts a systematic investigation of *ZoARF* genes by analyzing gene structure, chromosome localization, promoter *cis*-acting elements, phylogenetic relationships, and collinearity within the *ZoARF* gene family. Furthermore, we systematically examined the expression profiles of *ZoARFs* across distinct ginger tissues, developmental stages, and stress treatments. These results offer valuable insights into the functional characteristics of *ZoARF* family genes, thereby informing genetic improvement efforts for ginger.

## 2. Results

### 2.1. Identification and Sequence Analysis of ZoARF Family Genes

Our study identified 26 *ARF* genes in the ginger genome through bioinformatics analyses, assigning them unique designations (*ZoARF#01* to *ZoARF#26*) based on chromosomal positions (Table 1). Using the ExPASy platform, we determined the physicochemical properties of ZoARF proteins, including amino acid count, molecular weight (MW), and isoelectric point (pI). These properties exhibited significant variation among the 26 identified genes. Coding sequence (CDS) lengths ranged from 1377 bp (*ZoARF#08*) to 3429 bp (*ZoARF#04*), resulting in protein sequences ranging from 458 (ZoARF#08) to 1142 (ZoARF#04) amino acids (aa). Corresponding MW values spanned 50.86 kDa (ZoARF#08) to 127.34 kDa (ZoARF#04), while pI values ranged from 5.49 (ZoARF#07) to 8.55 (ZoARF#08), with an average of 6.42. Most pI values (5.0–7.0) indicate weakly acidic proteins. ZoARF proteins were predicted to localize subcellularly, with 1 in the mitochondrion, 2 in the cytoskeleton, 1 in the endoplasmic reticulum, and 22 in the nucleus.

### 2.2. Phylogenetic Analysis of the ZoARF Gene Family

A phylogenetic analysis was conducted to investigate the connections between ARFs in multiple species and to gain insights into the potential functions of ZoARFs. A phylogenetic tree was created using ARF protein sequences from *Arabidopsis* and rice (Figure 1; Appendix A). The ARF gene family was divided into four major clades: Clade I comprised 12 *Arabidopsis*, 6 rice, and 7 ginger genes (*ZoARF#11/15/16/21/24/25/26*); Clade II contained 2, 4, and 6 genes (*ZoARF#02/03/06/07/10/19*), respectively; Clade III included 3, 6, and 4 genes (*ZoARF#13/14/17/23*); and Clade IV encompassed 5, 9, and 9 genes from these species (*ZoARF#01/04/05/08/09/12/18/20/22*).

### 2.3. Gene Structure and Motif Composition of the ZoARF Family

In order to gain a deeper insight into the conserved motifs found in the *ZoARF* protein as well as the structure of the *ZoARF* gene, an analysis was conducted on its functional domains and the intron-exon structure (Figure 2b). The results indicate that all *ZoARFs* have multiple exons. As shown in Figure 2b, exon number ranged from 2 to 18, with an average of 12 exons per *ZoARF* gene (Table 1). The characteristics of 26 ZoARF protein domain features were examined using the MEME online analysis tool. Ten distinct motifs (termed motifs 1–10) were identified in the 26 ZoARF proteins (Appendix A). Most *ZoARFs* exhibited the same motif order: motif 2, motif 5, motif 10, motif 1, motif 3, motif 6, motif 7, motif 4, motif 8, motif 9 (Figure 2c). The positional distribution of conserved motifs within the same subfamily exhibits similarities. For instance, motifs 8 and 9 were predominantly localized to ZoARF Clades I, III, and IV, but were absent from Clade II members. Overall, the pattern of these motifs remained consistent within the same phylogenetic group, further supporting the phylogenetic relationship and classification of *ZoARFs*.

Analysis of conserved domains in ZoARF protein sequences revealed that each sequence includes DNA-binding (DBD; B3) and auxin-responsive (Auxin_resp) domains (Figure 2d). Among these, 14 ARF proteins contain Aux/IAA domains, whereas 12 lack these domains. This suggests that CTD-truncated ZoARFs may contribute to the regulation of plant biological processes through pathways independent of auxin.

### 2.4. Chromosomal Localization of ZoARF Family Genes

According to the physical locations of genes in the ginger genome, the positions of the *ARF* genes on the chromosomes were illustrated (Figure 3). It was found that the 26 *ZoARF* genes are distributed unevenly among nine of the eleven chromosomes, with no genes detected on chromosomes 3 or 8 (Chr03 or Chr08). Chr10 harbored the highest number of *ARF* genes (*n* = 5), while Chr04 and Chr06 each contained only one. Chr01, Chr02, and Chr11 carried three *ARF* genes each, and Chr07 and Chr09 contained four genes (Figure 3; Table 1). It is interesting to note that *ZoARF#02* and *ZoARF#03* have very similar positions on Chr01, and *ZoARF#17* and *ZoARF#18* exhibit analogous positions on Chr09. This suggests that the *ZoARF* gene family may have undergone tandem duplication events during its evolution, likely contributing to the functional and genetic diversity observed within the family.

### 2.5. Analysis of cis-Regulatory Elements in the Promoter Region of the ZoARF Gene Family

A total of 594 *cis*-acting regulatory elements were identified in the promoter regions of *ZoARF* genes, categorized into four categories: light responsiveness, phytohormone responsiveness, stress responsiveness, and growth and development (Figure 4b). Light-responsive elements accounted for 42.83% of the total, including motifs such as G-box, Box4, GT1-motif, Sp1, TCT-motif, 3-AF1 binding site, Gap-box, and GATT-motif (Figure 4; Appendix A). Notably, the G-box was absent exclusively in *ZoARF#05*, *ZoARF#08*, *ZoARF#10*, *ZoARF#14*, and *ZoARF#19*. Phytohormone-responsive elements constituted 35.41%, encompassing TCA-element (salicylic acid), TGA-element (auxin), and AuxRR-core. The AuxRR-core motif was detected exclusively in *ZoARF#04* and *ZoARF*#08. The abscisic acid response element (ABRE) was present in the promoter regions of all *ZoARF* genes except *ZoARF#14* and *ZoARF#19*. Gibberellin response elements comprised P-box, GARE-motif, and TATC-box, with the TATC-box restricted to *ZoARF#08*, *ZoARF#17*, and *ZoARF#23*. Jasmonic acid response elements (CGTCA-motif and TGACG-motif) were detected in all *ZoARF* genes except for *ZoARF#01*, *ZoARF#05*, *ZoARF#13*, *ZoARF#17*, *ZoARF#18*, and *ZoARF#19*. Stress-responsive elements (16.70%) included ARE (anaerobic induction detected in 20 *ZoARFs*, excluding *ZoARF#02*, *ZoARF#06*, *ZoARF#07*, *ZoARF#08*, *ZoARF#09*, and *ZoARF#19*), WUN-motif (injury response; only in *ZoARF#05* and *ZoARF#16*), LTR (low-temperature response; 13 *ZoARFs*), TC-rich repeats (defense and stress; 11 *ZoARFs*), and MBS (drought response; 10 *ZoARFs*). Growth and development-related elements (5.06%) comprised CAT-box (meristem specificity; 15 *ZoARFs*), circadian (circadian control; *ZoARF#12*, *ZoARF#14*, and *ZoARF#19*), and RY-element (exclusively in *ZoARF#26*).

### 2.6. GO Annotation of ZoARF Protein Sequences

A Gene Ontology (GO) annotation analysis of *ZoARF* genes was performed to elucidate the functional roles of ARF proteins in different biological processes within ginger. The results demonstrated their potential involvement in biological processes, cellular components, and molecular functions (Appendix A). Analysis of biological processes revealed that most ZoARF proteins are associated with shoot system development, post-embryonic development, response to auxin, DNA-templated transcription, and RNA biosynthetic processes. In growth and development, six genes (*ZoARF#04/11/12/16/20/25*) were implicated in leaf system development, while six others (*ZoARF#07/08/09/11/16/25*) were linked to flower development. Additionally, a subset of genes (*ZoARF#04/07/08/09/11/12/16/20/25*) was annotated for shoot system development (Appendix A). Molecular processes of ARF proteins revealed that 9 out of 26 ZoARF proteins displayed DNA binding, sequence-specific DNA binding, DNA-binding transcription factor activity, transcription regulator activity, and nucleic acid binding. Furthermore, cellular component analysis confirmed that the majority of ZoARF protein localization occurs in the nucleus. This matches the outcomes observed in the previous subcellular localization.

### 2.7. Gene Duplication and Synteny Analysis of the ZoARF Gene Family

Gene duplication serves as a crucial mechanism for the generation of new functions and for the expansion of gene families. Genes related and located within 200 kb of each other on the same chromosome are referred to as tandem duplications, while those exceeding this distance are classified as fragment duplications [34]. To investigate the amplification mechanism of *ZoARFs*, we conducted an analysis of both fragment and tandem duplications in the ginger genome utilizing MCscan. The results showed that there were no observed cases of tandem duplication within the *ARF* gene family. However, we identified nine pairs of fragment duplications that were found on corresponding chromosomes (Figure 5). Notably, Chr01 contained the highest number of duplicated gene pairs, with three distinct fragment duplication pairs identified on this chromosome. In contrast, Chr03, Chr04, and Chr05 lack fragment duplication gene pairs. Among the *ZoARF* genes, seven pairs showed replication events, including *ZoARF#02*-*ZoARF#03*, *ZoARF#02*-*ZoARF#06*, *ZoARF#01*-*ZoARF#05*, *ZoARF#03*-*ZoARF#06*, *ZoARF#10*-*ZoARF#19*, *ZoARF#13*-*ZoARF#17*, and *ZoARF#15*-*ZoARF#24*, all located in duplicated regions of the ginger genome (Figure 5; Appendix A). In addition, replication events to *ZoARF* genes were discovered in other genes on Chr10 and Chr11. Notably, collinear gene pairs such as *ZoARF#02*-*ZoARF#03* and *ZoARF#02*-*ZoARF#06* exhibited one-to-many relationships, suggesting homology. The findings of the study indicate that fragment duplication is a significant factor in facilitating the expansion of the *ZoARF* gene family.

To investigate selective pressures acting on *ZoARF*, we calculated Ka/Ks ratios. The Ka/Ks values for all *ZoARF* gene pairs were less than 1 (Appendix A), suggesting that their evolution has been mainly influenced by purifying selection.

### 2.8. Evolutionary Analysis of ZoARF Genes

To further investigate the phylogenetic relationships of the *ZoARF* gene family in ginger, comparative collinear maps were constructed between ginger and four representative plant species: two monocots (banana and rice) and two dicots (*Arabidopsis* and potato). In Figure 6, the collinearity relationship of ARF genes in various species is presented. Ginger and banana had the most orthologous *ARF* pairs (36) of the analyzed species, with collinearity blocks primarily located on Chr01, Chr02, and Chr10, while rice showed three homologous pairs. The lack of gene collinearity between dicotyledonous plants (*Arabidopsis* and potatoes) and ginger *ARF* genes raises the possibility that there are distinct evolutionary branches for monocots and dicots (Figure 6; Appendix A). The results indicate that *ARF* genes in bananas are phylogenetically similar to *ZoARF* genes in ginger species due to the close relationship between bananas and ginger as monocotyledons.

### 2.9. Protein-Protein Interaction Network Analysis

A protein-protein interaction (PPI) network was constructed using STRING, and its topological features were analyzed at the network level. Our analysis demonstrated that 9 ZoARF proteins interacted with 9 other proteins, including five members of the IAA family, AUX1 (auxin transporter-like protein 1), BZR1 (brassinazole-resistant 1), PIF4 (phytochrome interacting factor 4), and CRL1 (crown rootless 1) (Figure 7; Appendix A).

AUX1 exhibited the highest connectivity, interacting with six ZoARF proteins (ZoARF#04/05/09/12/20/22). ZoARF#05, ZoARF#12, ZoARF#18, and ZoARF#22 were predicted to interact with BZR1, which regulates downstream genes in the brassinosteroid (BR) signaling pathway to modulate plant growth, development, and immune responses. BZR1 also mediates plant responses to environmental stimuli, including drought, salinity, and other abiotic stresses [35].

ZoARF#01/05/08/18/22 were predicted to interact with PIF4. It is not only involved in plant response to light signals but also related to phytohormone signaling, such as growth hormone (auxin), gibberellins, abscisic acid, and so on. In addition, PIF4 combines various environmental signals, including light and temperature, to regulate the adaptive growth of plants [36]. CRL1, a key regulator of crown root formation in rice. It encodes a LOB domain-containing transcription factor and is a target gene in the auxin signaling pathway [37]. CRL1 acts as a positive regulator of crown and lateral root development, with ARF directly controlling its expression in the auxin signaling pathway. The network predicts interactions between CRL1 and ZoARF#04, ZoARF#14, and ZoARF#20, suggesting these genes play essential roles in crown and lateral root development.

### 2.10. Expression Profiling of Ginger ZoARF Genes in Different Tissues

Previous studies have established the significant role of ARF genes in plant growth and development. Building on this foundation, we investigated the expression of *ZoARF* genes in ginger across various tissues using transcriptome data from 12 different tissues (Figure 8a; Appendix A). An RNA-seq-derived expression heat map revealed notable variations in *ZoARF* gene expression. Specially, four *ZoARF* genes (*ZoARF#06*, *ZoARF#13, ZoARF#14* and *ZoARF#23*) were preferentially expressed in roots, three in young flower (*ZoARF#05*, *ZoARF#12*, *ZoARF#26*), five in mature flowers (*ZoARF#04*, *ZoARF#17*, *ZoARF#22* and *ZoARF#24*), three in flower petiole (*ZoARF#02*, *ZoARF#03*, and *ZoARF#09*), two in basal stem (*ZoARF#01* and *ZoARF#20*), two in shoot apical bud (*ZoARF#01* and *ZoARF#24*), one in leaves (*ZoARF#15*) exhibited relatively higher expression levels than other *ZoARFs*. Various patterns of *ZoARF* expression exhibited distinct trends at different stages of development. For instance, during the rhizome development stages, the expression levels of *ZoARF#02*, *ZoARF#08*, and *ZoARF#16* gradually increased while *ZoARF#12* gradually declined. These results imply that the *ZoARF* gene might be linked to the regulation of ginger root and flower growth and development.

Notably, *ZoARF#07*, *ZoARF#18*, *ZoARF#21*, and *ZoARF#25* had minimal expression levels or were undetectable in different tissues and organs. These genes are hypothesized to potentially be pseudogenes or to have distinct expression patterns in time or space that are not yet captured by the available data. We conducted a qRT-PCR analysis to examine the expression of five randomly selected genes in 12 diverse tissues. In summary, it was observed that *ZoARF#05* was expressed more in young flowers than in any other tissue, while *ZoARF#04* showed higher expression levels in mature inflorescences. Moreover, *ZoARF#13* expression was higher in roots than in other analyzed tissues (Figure 8b). This finding aligns with the RNA-seq data.

### 2.11. Expression Profiles of ZoARF Genes Under Abiotic Stress Conditions

RNA-seq data were analyzed to assess *ZoARF* gene expression under various stress conditions, including ABA, cold, heat, drought, and salt. Significant alterations in the expression levels of *ZoARFs* were noted under cold treatment conditions, including *ZoARF#05/06/11/12/23/24* as compared to the control (CK), with their levels being altered by more than 2-fold, suggesting their involvement in ginger’s temperature response. During drought stress, *ZoARF#01/03/08/12* was upregulated, with *ZoARF#03* and *ZoARF#08* increasing over 3-fold. Under conditions of salt stress, significant upregulation was observed in the expression of *ZoARF#01/05/09/12/23/25* in comparison to the control. Notably the expression levels of *ZoARF#05* and *ZoARF#09* were elevated by more than fourfold. Conversely, during heat stress, the majority of the *ZoARF* genes were downregulated, particularly *ZoARF#02/08/09/14/22/23*, with *ZoARF#02*, *ZoARF#09*, and *ZoARF#23* exhibiting more than a 4-fold reduction in expression levels. ABA treatment induced eight *ZoARF* genes, specifically *ZoARF#03/05/06/08/09/12/23/25* (Figure 9a; Appendix A), indicating their potential involvement in ABA-mediated stress-responsive regulatory pathways.

In contrast, drought, low temperature, ABA, and salt treatments downregulated 7, 8, 4, and 2 *ZoARF* genes, respectively. *ZoARF#02/09/10/17/19* were downregulated by both low-temperature and drought, while *ZoARF#02* and *ZoARF#19* were downregulated by both low-temperature, drought, ABA, and salt treatments. Additionally, qRT-PCR was employed to assess the expression of 6 randomly selected *ZoARF* genes under various stress conditions, including heat, low temperature, drought, salt, and ABA. These genes displayed significant upregulation at distinct time points post-exposure (Figure 9b), aligning with RNA-seq data from the 12 h treatment period. For instance, under salt stress, *ZoARF#05* expression rose progressively, attaining a fourfold increase relative to the control at 12 h and peaking at 24 h. It is significant to note that these genes exhibited peak expression at specific time points under various stress conditions, with notable differences in expression across different time intervals. Notably, during cold treatment, the expression of *ZoARF#25* demonstrated a declining trend (Figure 9b). Most genes responded more rapidly to low-temperature stress compared to other stress conditions, achieving their peak expression levels within 1 to 6 h post-treatment.

## 3. Discussion

Auxins are crucial regulators in plants. As transcription factors, *ARFs* mediate signaling pathways involved in auxin response, thereby influencing plant growth and development [7]. Extensive characterization of *ARF* gene families in species such as *A. thaliana* [13], *O. sativa* [14], and *Z. mays* [15] has demonstrated their essential roles in regulating organogenesis, including root, leaf, and floral development, as well as modulating plant responses to environmental stresses [27]. Currently, there is no comprehensive study of *ARF* genes in ginger. As the nutritional and medicinal properties of ginger continue to be investigated, its economic value is also on the rise. To understand the impact of *ARF* on specific auxin responses in ginger, we conducted the first comprehensive analysis of *ARF* family genes in this species.

Gene structure is typically conserved during evolution. Exon-intron structure plays a critical role in gene evolution and functional diversification [38]. Analysis of gene structure revealed that *ZoARF* genes comprise 2–18 exons (Figure 2b). While the number of exons varies significantly among different subfamilies, it aligns with the range of *ARF* gene exons observed in *Arabidopsis* [13], rice [14], apple [39], and other species, indicating that *ARF* is relatively conserved throughout species evolution. Furthermore, ARF proteins within the same phylogenetic clade exhibit shared evolutionary origins and possess conserved motifs linked to their functions [40]. Analysis of the conserved domains of ZoARF proteins revealed that each ZoARF contains a B3 DNA-binding domain and Auxin_resp domain. Additionally, twelve ZoARFs were identified as lacking the CTD in this study. The proportion of CTD-truncated ZoARFs (46.15%) in ginger is significantly higher than that in tomato (22.73%) [41], rice (24%) [14], and maize (30.56%) [42], yet lower than that of *Medicago truncatula* (54.17%) [10] and *Fagopyrum dibotrys* (57.69%) [43]. This indicates that ARF proteins lacking the CTD are involved in the regulation of growth and development in plants, a regulatory mechanism that likely operates in ginger as well [10].

In this study, we identified the 26 *ARF* genes from the ginger genome. The number of *ZoARF* genes (26) exceeds those in *Arabidopsis* (23) [13] and barley (23) [44], is lower than in maize (31) [15], and is comparable to rice (25) [14], sorghum (25) [45], and coix (26) [46]. The ginger genome (1.53 Gb) [30] is significantly larger than *Arabidopsis* (135 Mb) [47], rice (430 Mb) [48], sorghum (730 Mb) [49], and coix (1.2 Gb) [46], but smaller than maize (2.16 Gb) [50] and barley (5.1 Gb) [51]. These findings suggest that *ARF* gene family size is influenced by gene duplication rather than genome size [43,52]. The *ARF* gene family expands in crops through tandem or segmental gene duplication mechanisms [53,54].

Tandem duplications have been documented in the *ARF* gene family across various species, including Arabidopsis [55] and rice [14]. In contrast, the present study demonstrates that the *ZoARF* gene in ginger undergoes only segmental duplication events, with no evidence of tandem duplications. This suggests that gene duplication events may significantly influence the evolution and functionality of the *ZoARF* gene [52]. Collinearity analysis identified the formation of nine *ZoARF* pairs through segmental duplications, with Ka/Ks values for these pairs being less than 1 (Figure 5; Appendix A) [56]. This finding suggests that *ZoARF* genes have been subject to negative purifying selection to preserve their function, a pattern similarly observed in *Fagopyrum dibotrys* [43] and *Coix lacryma-jobi* [46].

Gene duplication not only contributes to gene family expansion but also promotes functional diversification. Notably, collinear gene pairs such as *ZoARF#02*-*ZoARF#03* and *ZoARF#02*-*ZoARF#06* display one-to-many relationships, indicating that many *ZoARFs* are homologous genes. Analyzing gene expression profiles can provide preliminary insights into gene function. For instance, both roots and flower petioles showed substantial expression levels of *ZoARF#02*, *ZoARF#03*, and *ZoARF#06* (Figure 8; Appendix A). *ZoARF#02* and *ZoARF#03* showed predominant expression in flower petioles, whereas *ZoARF#06* exhibited the highest expression in roots. The genes *ZoARF#03* and *ZoARF#06* contain motif-3 among other motifs (motif-2, motif-5, motif-10, motif-1, motif-3, motif-6, motif-7, motif-4), while *ZoARF#02* lacks motif-3, but includes the other motifs (motif-2, motif-5, motif-10, motif-1, motif-6, motif-7, motif-4) (Figure 2c). This observation suggests that alteration in motif composition, potentially due to segmental duplication, may contribute to functional divergence.

Tissue-specific expression profiles revealed that a majority of Clade I (Figure 1) genes (4/7, 57.1%), Clade II genes (4/6, 66.7%), and Clade III genes (4/4, 100%) were expressed across all examined tissues. Furthermore, Clade IV genes were expressed in 8 out of 9 tissues (88.9%) (Figure 8a). It is noteworthy that genes with high homology within the same branch of the phylogenetic tree may perform similar functions. The phylogenetic analysis further reveals that *ZoARF#08*, *AtARF6*, and *AtARF8* exhibit significant homology. In *A. thaliana*, the roles of *AtARF6* and *AtARF8* [23,57] in jasmonic acid (JA) synthesis and the regulation of floral organ maturation provide a potential framework for elucidating the function of *ZoARF#08*. Notably, *ZoARF#08* harbors the TGACG and CGTCA motif within its *cis*-acting elements, suggesting a possible involvement in the regulation of jasmonic acid synthesis, which warrants further investigation (Figure 4; Appendix A).

It is widely recognized that ginger exhibits infrequent flowering, often blooming only once per decade. Despite this, the mechanisms underlying ginger’s flowering process remain poorly understood, highlighting the importance and potential benefits of research in this area. Our study reveals that *ZoARF* is significantly expressed during flower development (Figure 8a). Specifically, the expression levels of *ZoARF#04*, *ZoARF#05*, *ZoARF#12*, and *ZoARF#22* are elevated in young and mature flowers compared to buds. These findings indicate that the *ZoARF* gene may play a role in the regulation of ginger flower growth and development. In addition, ZoARF#04/05/12/22 is classified as the Clade IV subfamily. AtARF6 and AtARF8, also members of Clade IV, play critical roles in flower maturation by regulating the development of floral organs, which is intrinsically linked to the flowering process [57,58]. We hypothesize that the homologous *ZoARF* genes may exhibit similar functions due to their shared evolutionary lineage within the phylogenetic tree. Analysis of *cis*-acting elements reveals that the promoter regions of *ZoARF#04/05/12/22* are enriched with light-responsive elements, suggesting their potential involvement in flower development during plant growth (Figure 4; Appendix A). GO analysis of ZoARF#04 and ZoARF#12 further supports this, as it reveals their participation in biological processes related to flower development, corroborating our tissue expression findings (Appendix A). Additionally, the *AtARF* gene family may interact with other hormone signaling pathways, such as those involving gibberellin and ethylene, within the intricate regulatory network governing plant flowering, collectively influencing reproductive growth [26]. To fully understand the role of the *ZoARF* gene family in flowering development, it is essential to consider the interplay with other flowering regulatory genes and environmental factors.

As a crop of significant economic value, ginger’s economic importance is largely dependent on rhizome expansion. In ginger, the expression levels of *ZoARF#06*, *ZoARF#13*, *ZoARF#14,* and *ZoARF#23* are notably higher in roots compared to leaves and other vegetative organs (Figure 8a; Appendix A), indicating a potential role in root development. Phylogenetic analysis reveals that *ZoARF#13*, *ZoARF#14*, and *ZoARF#23*, which exhibit high expression in roots, are clustered together with AtARF10 and AtARF16 within Clade III (Figure 1). In *Arabidopsis*, *AtARF10* and *AtARF16* have been shown to be crucial for root development [59], suggesting that *ZoARF#13* and *ZoARF#14* may be functionally conserved in ginger. Additionally, *AtARF6* has been identified as a positive regulator of adventitious roots development in *Arabidopsis* [60], implying that *ZoARF#06* may similarly act as a positive regulator of root development. This evidence suggests that certain *ZoARFs* in ginger may perform analogous functions to their *Arabidopsis* counterparts.

Gingerols and curcuminoids are recognized as the primary bioactive constituents in ginger [27]. The rhizomes of ginger contain a higher concentration of these compounds compared to other plant parts. *AtARF5* serves as a key transcriptional activator within the auxin signaling pathway, playing a significant role in embryonic development and vascular tissue formation [22]. Phylogenetic analyses have identified *ZoARF#04* as an ortholog of *AtARF5*. Our research indicates that *ZoARF#04* exhibits peak expression in the young nodes (specifically the 1st inter-node) during rhizome development (Figure 8a; Appendix A), implicating its potential involvement in rhizome elongation. Notably, our investigation of the protein interaction network revealed an interaction between *ZoARF#04* and CRL1. These findings lead us to hypothesize that *ZoARF#04* may play a role in rhizome formation and is potentially linked to the biosynthesis of curcumin and gingerol compounds.

According to Runjie Diao et al., the BZR1-ARF6-PIF4 (PAP) module, which comprises BZR1, ARF6, and PIF4, physically interacts to form an integrated signaling network involving light, brassinosteroids (BR), and auxin, thereby regulating plant growth and development [61]. In an intriguing finding, ZoARF#05, ZoARF#18, and ZoARF#22, which are phylogenetically aligned with AtARF6, were identified within the protein interaction network that includes BZR1, PIF4, and ZoARF proteins. This observation implies a potential collaborative role of these genes, alongside BZR1 and PIF4, in regulating ginger growth and development (Figure 7; Appendix A).

Our analysis of the *ZoARF* gene against existing transcriptome data led to preliminary identification of genes exhibiting significant differential expression under conditions of drought, salt, heat, cold, and ABA stress, suggesting their candidacy as ARF genes responsive to abiotic stress. It has been reported that most *OsARF* genes are repressed by heat and cold stress, whereas *OsARF11*, *OsARF13*, and *OsARF16* are activated by heat stress, and *OsARF4*, *OsARF14*, *OsARF18*, and *OsARF19* are induced by cold stress [62]. Meanwhile, their sorghum homologs, *SbARF6*, *SbARF24*, and *SbARF25*, show significant induction under cold stress, while *SbARF16* and *SbARF22* are markedly induced by heat stress [45]. In this study, we found that seven *ZoARF* genes (*ZoARF#06/11/14/15/23/24/26*) were upregulated, and five *ZoARF* genes (*ZoARF#04/09/10/18/19*) were downregulated in ginger subjected to cold stress. In contrast to the responses observed under cold stress, the majority of *ZoARF* genes showed significant downregulation when subjected to heat stress conditions (Figure 9). A comparative analysis indicated that *ARF* genes in rice, sorghum, and ginger were activated by both heat and cold stress, implying that auxin may play a role in mediating plant responses to temperature variations. Zhou et al. have shown that *OsARF2/4/10/14/16/18/19/22/23* exhibit sensitivity to drought stress [63]. Furthermore, under drought conditions, 33 out of 51 *ARF* genes in soybean exhibited upregulation of their expression levels. In this study, *ZoARF#03/08/13* was significantly induced by drought stress, suggesting that these *ZoARFs* play a critical role in drought stress response.

Notably, *ZoARF#07/08/10/20/21/25* showed higher induction in the presence of abscisic acid (ABA) compared to conditions of drought, low temperature, heat, and salt stress. In *Arabidopsis*, ABA has been shown to induce the expression of *AtARF2*, and overexpression confers enhanced resistance to ABA relative to the wild type [64]. Additionally, phylogenetic analysis revealed that *ZoARF#21* and *ZoARF#25* are grouped within subfamily I alongside *AtARF2*. It is posited that *ZoARFs* play a crucial role in the ABA stress response. In rice, OsARF11 exhibits unique expression patterns when subjected to salt stress, suggesting its involvement in adaptation to such conditions [65]. In this study, *ZoARF01/04/05/09/18*, which are members of the same subfamily, demonstrated elevated expression levels under salt stress conditions. Notably, the promoters of these *ZoARF* genes contain TC-rich repeats, MBS, ARE, and CGTCA motifs, all of which are recognized for their role in enhancing plant stress resistance. Previous research has indicated that salt stress leads to the upregulation of genes encoding auxin response proteins in *Arabidopsis thaliana* [66]. It is hypothesized that analogous biological processes may occur in ginger. However, further validation evidence is required to substantiate this hypothesis.

## 4. Materials and Methods

### 4.1. Identification Analysis of ZoARFs

According to the genomic information obtained from our investigation of the ginger genome. Using BLASTP, candidate genes having an e-value of ≤e^−10^ and a score value of ≥100 were chosen. The Pfam protein family database provided the ARF domain HMM profile (PF06507) (http://pfam.xfam.org/, accessed on 14 November 2024). To confirm each potential candidate’s ARF gene, a manual evaluation was carried out using the Pfam database.

### 4.2. Analysis of the Physicochemical Properties of the ZoARFs Gene Family

The physicochemical characteristics of individual ARF genes, including metrics such as protein length, exon-intron distribution, isoelectric point (pI), and molecular weight (MW), were ascertained through the utilization of the ExPASy online platform (http://web.expasy.org/compute_pi/, accessed on 24 November 2024). The subcellular localization of each ZoARF was subsequently identified through the utilization of WoLF PSORT (https://wolfpsort.hgc.jp/, accessed on 25 November 2024) [67].

### 4.3. Phylogenetic Analysis, Gene Structure, and Motif Composition of the ZoARFs

The protein sequences of the ZoARF gene family were analyzed by comparing them with the *ARF* gene families of *Arabidopsis thaliana* and *Oryza sativa* retrieved from the PlantTFDB database (https://planttfdb.gao-lab.org/, accessed on 27 November 2024) (Appendix A) [68], using MEGAv11.0.13 software. Furthermore, a maximum likelihood (ML) analysis was carried out using FastTree 2.2, applying the JTT+G substitution model and including 1000 non-parametric bootstrap replicates. The generated tree was displayed using the online software iTOL (https://itol.embl.de/, accessed on 30 November 2024) [69]. The conserved domains of ZoARFs were predicted using the NCBI-CDD online resource (https://www.ncbi.nlm.nih.gov/cdd/, accessed on 5 December 2024). Exon-intron sequences were extracted from ZoARF genes using the genome annotation gff3 file, followed by an analysis of their structure and diagram construction using TBtools v2.052 [70]. Analysis of motifs in the ZoARF protein sequences was conducted using the MEME Suite [71], with a maximum of 10 motifs specified.

### 4.4. Analysis of cis-Acting Elements

The *cis*-acting elements within the promoter regions of ginger *ZoARF* genes were investigated by analyzing the 2000 bp upstream sequences of *ZoARFs* using the Plant-CARE online database [72]. Subsequently, the *cis*-acting elements were categorized and tallied, and presented as histograms using SigmaPlot v12.5 software. Furthermore, heatmaps depicting distinct categories of *cis*-acting elements were constructed using TBtools v2.052.

### 4.5. Chromosomal Distribution, Genomic Duplication, and Ka/Ks Ratios of ZoARF Genes

To determine the positions of *ZoARF* genes on the chromosomes of ginger, the starting and ending coordinates of their coding sequences (CDS) were extracted from the ginger genomic data. Each *ZoARF* gene was subsequently mapped to its respective chromosome in ginger based on these physical coordinates. Protein sequences for *ARF* genes from *A. thaliana*, *S. tuberosum*, *O. sativa*, and *M. acuminata* were retrieved from the UniProt database (https://www.uniprot.org/, accessed on 20 December 2024). Through BLAST analysis [73], homologous genes were identified. Syntenic relationships were examined using MCScanX [74]. To evaluate the selective pressure and divergence time between pairs of *ZoARF* genes, Ka/Ks ratios were computed. Furthermore, TBtools v2.052 [70] facilitated the collinearity analysis of the ginger genome alongside gene annotation files. The Ka/Ks calculator was used to calculate the rates of non-synonymous substitutions (Ka), synonymous substitutions (Ks), and the selection pressure (Ka/Ks) for paralogous gene pairs [75].

### 4.6. Protein Interaction Analysis

The online interacting protein search site STRING was used to predict interactions between ginger ZoARF proteins (http://string-db.org/, accessed on 30 December 2024) [76], and protein-protein interaction networks were mapped and improved using Cytoscape v3.10.1 (http://www.cytoscape.org/) [77].

### 4.7. Plant Materials

The study examined the ginger cultivar “Laiwu No. 2”, provided by Shandong Second Academy of Agricultural Sciences, China. The seed ginger, characterized by healthy buds, was divided into approximately 30 g pieces, each with 2–3 buds, and planted in 40 cm by 20 cm pots filled with a 6:3:1 mix of peat, garden soil, and perlite. The seedlings are grown in a greenhouse with 16/8 h of light/darkness daily. The expression of the *ZoARF* genes was examined in seedlings approximately 6 months old, encompassing various plant parts such as leaves (particularly the third spot from the root tip towards the base of the stem), roots, leaf buds, rhizome buds, flower buds, mature flowers, the base of the stem, flower stalk, and the rhizome internode-1st, -2nd, and -3rd. Two-month-old seedlings were subjected to salt and drought stress using 200 mM NaCl and 15% PEG6000 solutions to study the gene’s response to abiotic stress. Additionally, a 0.1 mM ABA solution was applied to the ginger leaves. At 40 °C and 4 °C, respectively, the heat and cold stressors were addressed. After subjecting the plants to salt, low temperature, and drought treatments, leaf samples were collected at different intervals: 0, 1, 3, 6, 12, 24, and 48 h. Each response was given three times. At specific time intervals of 0, 1, 3, 6, 12, and 24 h, leaf samples were collected for the purpose of heat treatment (Appendix A). The collected samples were immediately frozen in liquid nitrogen and stored at –80 °C.

### 4.8. Expression Analysis of ZoARF Genes by RNA-Seq and qRT-PCR

RNA-seq was conducted on samples collected 12 h after subjecting them to treatments involving abscisic acid (ABA), heat, cold, salt, and drought. The FPKM value of gene expression in RNA-Seq data was analyzed. The FPKM value is converted to the log_2_ (FPKM+1) value using TBtools v2.052 software, and a thermal expression heatmap is created specifically for the *ZoARF* gene. The samples were processed for RNA extraction employing the Plant RNeasy Mini Kit (Qiagen, Hilden, Germany, Code number: 74904), followed by cDNA synthesis through the PrimeScript RT reagent Kit with gDNA Eraser (Takara, Dalian, China; Code Number: RR047A). The TP9500 Thermal Cycler Dice Real Time System (Takara) was used to conduct quantitative reverse transcription polymerase chain reaction (RT-qPCR) with TB Green Fast qPCR Mix (Takara, Code Number: RR430A). The reaction setup had a volume of 10, which included 5 μL of SYBR mix, 0.4 μL of primer mix (10 μM), 0.5 μL of cDNA template, and 4.1 μL of ddH_2_O. The settings for qRT-PCR amplification were the following: 95 °C for 1 min, then 40 cycles of 10 s at 95 °C and 30 s at 60 °C. As an internal reference, use the *ZoTUB2* gene. Three biological duplicate experiments were carried out. The 2^−ΔΔCT^ method was used to measure relative expression levels of *ZoARFs* [78]. The software DNAman 8.1.4.87 was utilized to design qRT-PCR primers (Appendix A).

## 5. Conclusions

Ginger is an important economic crop with both medicinal and edible values. In this study, 26 *ZoARF* genes were systematically identified from the *Zingiber officinale* genome. Phylogenetic analysis classified these *ZoARFs* into four distinct subgroups, each characterized by a similar gene structure and motif composition. The results demonstrate that these genes are distributed non-uniformly across the 11 chromosomes. Subcellular localization prediction analysis indicated that the ZoARF protein predominantly resides in the cell nucleus. Evolutionary analysis suggested a close relationship between ZoARFs and banana ARF genes, implying a common ancestral origin. Collinearity analysis revealed that segmental duplication events have resulted in the formation of nine pairs of *ZoARF* genes. Furthermore, an examination of *cis*-acting elements within the promoter region suggests that the *ZoARF* genes play a crucial role in ginger growth and development, particularly in response to hormonal and abiotic stressors. Based on the PPI network, the construct containing 9 ZoARF proteins (ZoARF#01/04/05/08/09/12/18/20/22), 5 IAA proteins, BZR1, PIF4, and CRL1 had significant interactions. The expression patterns of *ZoARF* across various tissues and organs underscore its vital roles in the growth and developmental processes of ginger. Notably, *ZoARF#04/05/12/22* was discovered to influence blossom growth in ginger, whereas *ZoARF#06/13/14/23* was associated with root development. Furthermore, analysis of RNA-Seq datasets reveals that the *ZoARF* gene responds to abiotic stress. It is noteworthy that 4 *ZoARF* genes were induced by cold, 4 by drought, 8 by ABA, and 5 by salt. This research provides deeper insights into the diversity of ARF genes in ginger and establishes a theoretical framework for future investigations into their functional roles.

## Figures and Tables

**Figure 1 ijms-26-08412-f001:**
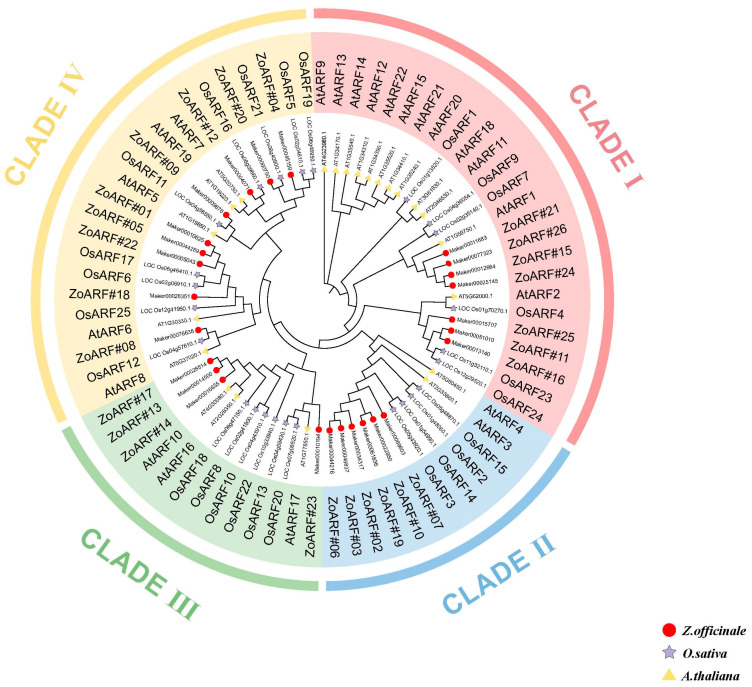
Phylogenetic analysis of ARF protein in *A. thaliana*, *O. sativa*, and *Z*. *officinale.* This analysis employed ARF protein amino acid sequences from these species and was constructed using the maximum likelihood (ML) method in MEGA11. The numbers at the branches indicate the confidence values obtained from the 1000 bootstrap tests. Roman numerals I–IV represent different ARF groups.

**Figure 2 ijms-26-08412-f002:**
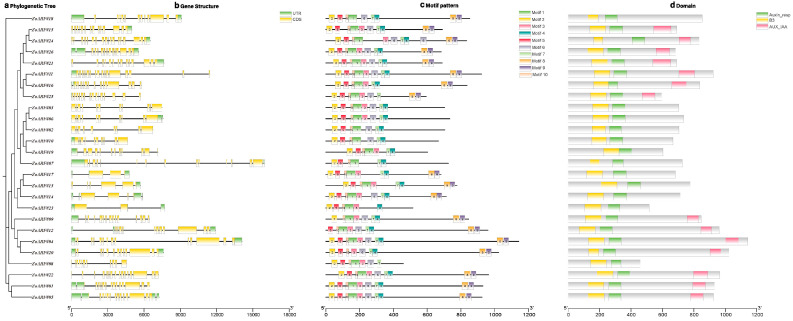
Phylogenetic relationships, gene structures, and conserved protein motifs of *ARF* genes in ginger. (**a**) Phylogenetic tree of *ARF* genes in ginger. (**b**). The exon-intron structure of ginger *ARF* genes. Green boxes represent untranslated regions (UTRs), yellow boxes denote exons, and black lines indicate introns. (**c**) Motif composition of ginger ARF proteins. Ten conserved motifs (1–10) are illustrated as distinct colored boxes. (**d**) Conserved domains in ZoARF proteins: green boxes represent B3 structural domains, yellow boxes represent Auxin_resp structural domains, and pink boxes represent Aux/IAA domains.

**Figure 3 ijms-26-08412-f003:**
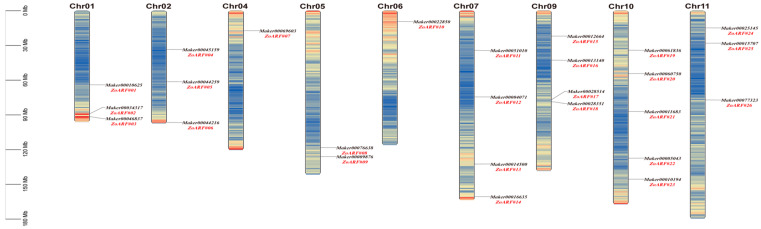
The Chromosomal distribution of *ZoARF* genes. The scale on the left indicates chromosome length in megabases (Mb). Chromosome numbers (excluding Chr03 and Chr08) are labeled at the top of each chromosome. *ZoARF* gene numbers are marked in red to the right of each chromosome.

**Figure 4 ijms-26-08412-f004:**
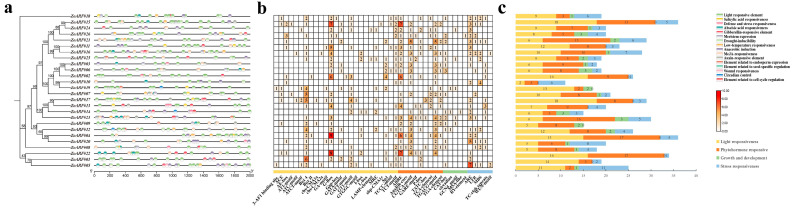
*cis*-regulatory elements of the *ZoARF* gene promoters. (**a**) *cis*-acting elements in the promoter regions of *ZoARF* genes in ginger. Distinct *cis*-acting elements are represented by colored diagrams. (**b**) *ZoARF* promoters are represented in a heatmap, with red indicating a higher count of *cis*-elements and orange indicating a lower count. (**c**) Histograms display the proportion of *cis*-elements, using different colors to represent their functional categories.

**Figure 5 ijms-26-08412-f005:**
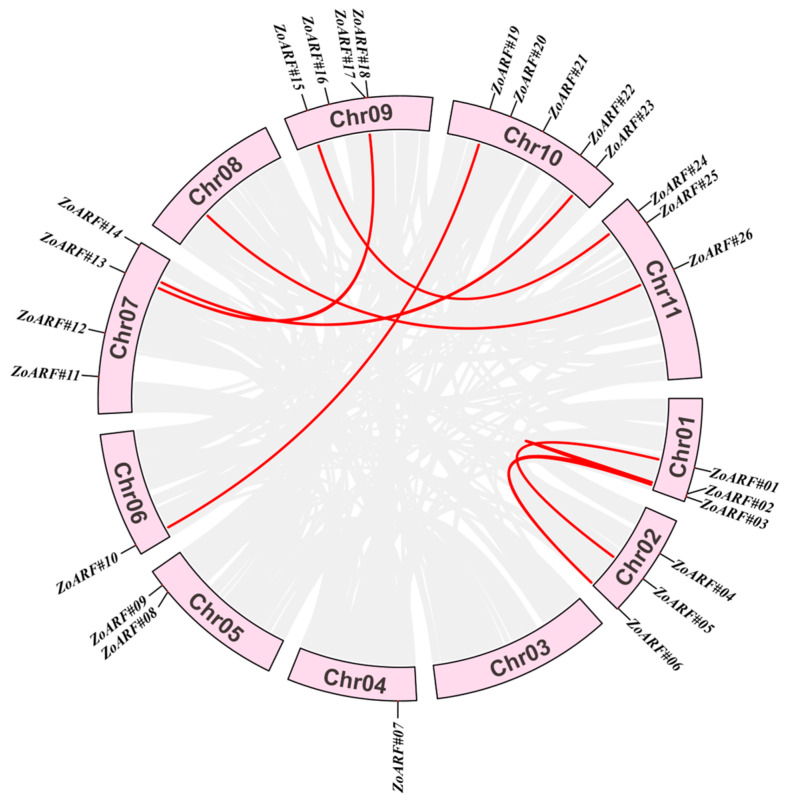
Schematic diagram of *ZoARFs* collinearity in ginger. The gray lines show the ginger genome’s collinear blocks, while the red lines denote pairs of duplicated *ARF* genes in ginger.

**Figure 6 ijms-26-08412-f006:**
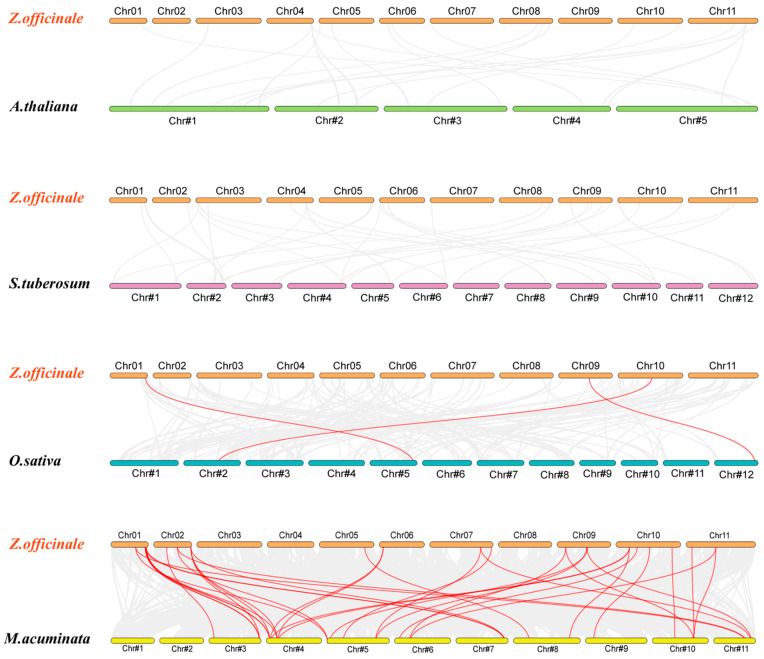
The collinear relationships of *ARF* genes were analyzed between *Zingiber officinale* (ginger) and four other plant species. *O*. *sativa* (rice), *A*. *thaliana* (*Arabidopsis*), *S*. *tuberosum* (potato), and *M*. *acuminata* (wild banana). Grey lines in the background illustrate collinear blocks between the ginger genome and the other plant genomes, whereas red lines indicate syntenic pairs of *ARF* genes.

**Figure 7 ijms-26-08412-f007:**
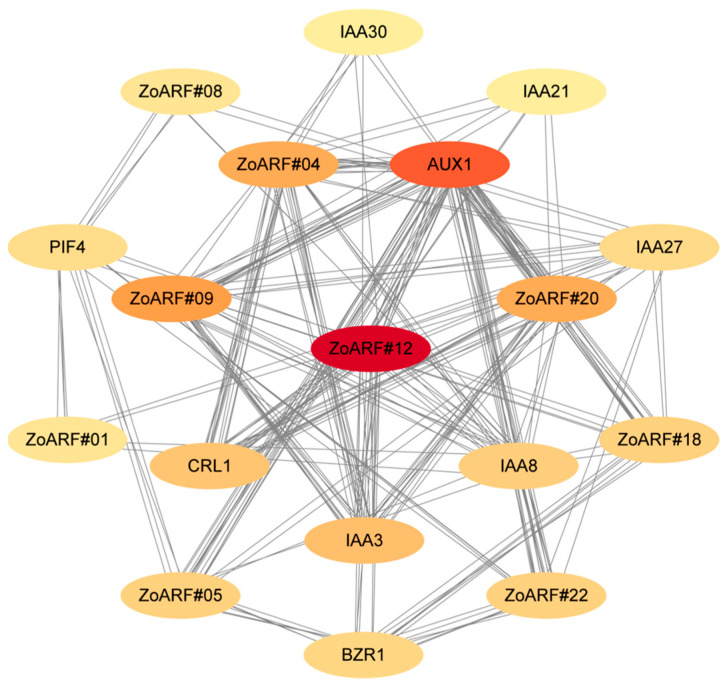
The ZoARF protein interaction network. The proteins with a higher degree of connection were represented by redder ovals.

**Figure 8 ijms-26-08412-f008:**
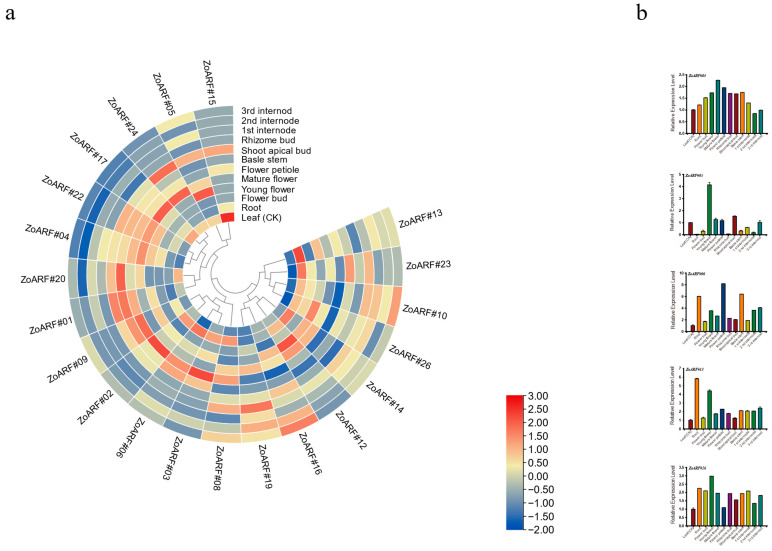
Expression profiling of the *ZoARF* gene in ginger. (**a**) RNA-seq was used to perform hierarchical clustering analysis of *ZoARF* gene expression levels in 12 distinct ginger tissues and developmental stages. (**b**) The expression of 5 *ARF* genes in 12 samples was analyzed using the qRT-PCR. Data normalization was performed using the TUB-2 gene, with vertical bars representing the standard deviation.

**Figure 9 ijms-26-08412-f009:**
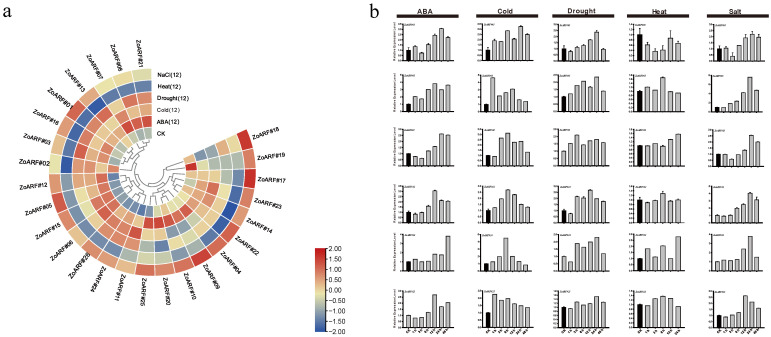
Expression profiles of *ZoARF* genes under various abiotic stress treatments were analyzed. (**a**) The expression of the *ZoARF* gene under diverse abiotic stressors was analyzed using RNA-seq. Relative expression of various *ZoARF* genes under ABA, cold, drought, heat, and salt conditions was measured 12 h after stress. (**b**) The qRT-PCR expression analysis data of ARF genes under abiotic stress were normalized to the *ZoTUB-2* gene, with the vertical bar graph representing the standard deviation.

**Table 1 ijms-26-08412-t001:** Basic information on ARF transcription factor in ginger.

Gene ID	Rename	Group	Chromsome	mRNA Start	mRNA End	Strand	No. of Exon	No. of Intron	CDS Length	Amino Acid Length	Isoelectric Point	Molecular Weight	Subcellular Localization
*Maker00034317*	*ZoARF#02*	II	Chr01	88858607	88865333	−	9	8	2115	704	6.09	76,431.35	nucleus
*Maker00010625*	*ZoARF#01*	IV	Chr01	63966592	63973047	−	14	13	2793	930	5.96	103,376.9	nucleus
*Maker00046837*	*ZoARF#03*	II	Chr01	91467289	91474763	+	11	10	2112	703	6.49	77,384.55	nucleus
*Maker00044216*	*ZoARF#06*	II	Chr02	96517654	96525217	−	10	9	2205	734	6.36	80,760.08	nucleus
*Maker00045159*	*ZoARF#04*	IV	Chr02	33489395	33503493	+	14	13	3429	1142	6.29	127,342.96	nucleus
*Maker00044259*	*ZoARF#05*	IV	Chr02	61251400	61258631	−	14	13	2775	924	6.21	102,663.2	nucleus
*Maker00069603*	*ZoARF#07*	II	Chr04	17191057	17206998	+	14	13	1680	559	5.49	81,066.21	nucleus
*Maker00009876*	*ZoARF#09*	IV	Chr05	125989767	125996191	+	15	14	2538	845	5.54	94,659.43	nucleus
*Maker00076638*	*ZoARF#08*	IV	Chr05	118203804	118208349	−	11	10	1377	458	8.55	50,857.31	cytoskeleton
*Maker00022850*	*ZoARF#10*	II	Chr06	9308738	9313381	−	10	9	2001	666	6.83	73,121.55	nucleus
*Maker00016635*	*ZoARF#14*	III	Chr07	160616034	160621927	−	4	3	2142	713	7.53	78,081.88	nucleus
*Maker00051010*	*ZoARF#11*	I	Chr07	34201967	34213404	+	16	15	2766	921	6.21	102,241.46	nucleus
*Maker00014500*	*ZoARF#13*	III	Chr07	132442912	132448595	+	6	5	2331	776	7.03	86,413.33	nucleus
*Maker00004071*	*ZoARF#12*	IV	Chr07	74447622	74459523	−	11	10	2883	960	6.07	107,389.35	nucleus
*Maker00028351*	*ZoARF#18*	IV	Chr09	78769700	78778791	−	12	11	2559	852	5.81	94,688.59	nucleus
*Maker00013140*	*ZoARF#16*	I	Chr09	42682558	42688310	−	14	13	2511	836	6.04	93,016.74	nucleus
*Maker00012664*	*ZoARF#15*	I	Chr09	21899860	21904846	+	14	13	2073	690	6.36	77,079.79	Endoplasmic reticulum
*Maker00028514*	*ZoARF#17*	III	Chr09	76701982	76706759	+	3	2	2046	681	6.97	75,501.6	nucleus
*Maker00060750*	*ZoARF#20*	IV	Chr10	54271985	54279602	−	13	12	3072	1023	6.04	114,496.96	nucleus
*Maker00010194*	*ZoARF#23*	III	Chr10	145551145	145558840	−	2	1	1545	514	6.53	56,971.54	cytoskeleton
*Maker00011683*	*ZoARF#21*	I	Chr10	87004599	87012232	−	14	13	2070	689	6.62	77,586.95	mitochondrion
*Maker00005043*	*ZoARF#22*	IV	Chr10	127366956	127374139	−	16	15	2892	963	6.18	107,125.42	nucleus
*Maker00061836*	*ZoARF#19*	II	Chr10	34100555	34107690	+	12	11	1809	602	7.83	67,207	nucleus
*Maker00015707*	*ZoARF#25*	I	Chr11	28036173	28041877	+	15	14	1785	594	5.87	66,223.6	nucleus
*Maker00025145*	*ZoARF#24*	I	Chr11	14800662	14807162	+	18	17	2502	833	6.02	93,122.76	nucleus
*Maker00077323*	*ZoARF#26*	I	Chr11	77040284	77045821	−	14	13	2052	683	6.04	75,684.72	nucleus

## Data Availability

This study is an original research paper. All data used in this study are included in the main text and Appendix A of this paper. For any further questions, please contact the corresponding author of this article. The resulting transcriptome data have been archived in the NCBI Sequence Read Archive under accession number SRP476742.

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
