# Peer review of "Genome-Wide Analysis and Expression Profiles of Auxin Response Factors in Ginger (*Zingiber officinale* Roscoe)"

_ijms, 2025, doi:10.3390/ijms26178412_

Round 1

Reviewer 1 Report

Comments and Suggestions for Authors

The manuscript "Genome-Wide Analysis and Expression Profiles of Auxin Response Factors in Ginger (Zingiber officinale Roscoe)" is devoted to the identification and study of the ARF gene family in the ginger genome, genes associated with one of the main phytohormones - auxin. The manuscript is written in simple and understandable language, easy to read. The material is presented logically and consistently. What distinguishes this work from many other studies on the identification and analysis of certain genes in different plant species is that the data obtained using bioinformatics methods are supported by an experiment to assess the expression level of the identified genes during the development of ginger plants and during exposure to various stress factors. However, there are several comments on the work.

Major:

  1. How do the data from the two experiments described in sections 2.10 and 2.11 agree? At least one of the genes (ZoARF#25) shows a detectable expression level in qRT-PCR analysis under stress conditions. Thus, the phrase on lines 312-314 contradicts the data presented in the next section. Perhaps the list of genes and a summary of which genes can be classified as pseudogenes should be given at the very end of the Results section, based on the entire array of data obtained?
  2. Why were the genes for qRT-PCR chosen randomly, and not based on some analysis and an attempt to identify the most active/promising/associated with some process of interest? Not based on the synthesis of all the data obtained, which is given in the "Discussion" section?
  3. In the Conclusion section, it would be good to add a few suggestions regarding the prospects for further work, exploration of unresolved issues and approaches that can be used for this.

Minor:

  1. Lines 188-189 – apparently numbers 293-294 got here by mistake?
  2. Caption for Figure 6 - if the full Latin name has already been mentioned in the text, an abbreviated name can be used in the captions to the figures.
  3. In the Discussion and Materials and Methods sections – the same. After the first mention of the full Latin name in the text of the manuscript, the abbreviated one can be used.
  4. Line 339 – typo in the word “reduction”.
  5. Figure 9b – some unnecessary details are visible in the graph in the lower right corner.

Author Response

Dear reviewer,

We really appreciate to you for yourthe thoughtful review and constructive feedback provided by the reviewers. We agree with your suggestion and have incorporated their suggested changes into the manuscript. In order to increase the readability and quality of the article, the professional editing was performed for grammar and presentation. 

In this revised manuscript, modifications made in response to your suggestions are highlighted with a yellow background, while changes addressing errors or inappropriate English grammar and vocabulary are highlighted with a green background.

We make a point-by-point response to reviewers in the bottom of this letter.

We sincerely appreciate the time and effort invested by the reviewers in evaluating our manuscript. We look forward to any additional feedback or suggestions.

Best,

Hai-Tao Xing

xinght@cqwu.edu.cn 

Jul.30, 2025

Reviewer 1:

  1. How do the data from the two experiments described in sections 2.10 and 2.11 agree? At least one of the genes (ZoARF#25) shows a detectable expression level in qRT-PCR analysis under stress conditions. Thus, the phrase on lines 312-314 contradicts the data presented in the next section. Perhaps the list of genes and a summary of which genes can be classified as pseudogenes should be given at the very end of the Results section, based on the entire array of data obtained?

Response: Thanks for your valuable suggestion. For the RNA-seq sequencing data, we performed qRT-PCR to validate the reliability of the RNA-seq results by randomly selecting target genes. Concerning the ZoARF #25 gene, which was undetectable in tissue expression but present under stress conditions, implying that this gene demonstrates expression specificity, being expressed exclusively under particular induction conditions. Our expression analysis confirmed the detectability of all these genes, although some displayed variability under different induction scenarios. Consequently, we did not classify any genes as pseudogenes. We express our sincere gratitude for the reviewer's insightful suggestions.

  1. Why were the genes for qRT-PCR chosen randomly, and not based on some analysis and an attempt to identify the most active/promising/associated with some process of interest? Not based on the synthesis of all the data obtained, which is given in the "Discussion" section?

Response: We sincerely appreciate the reviewer's insightful comment regarding the selection of genes for qRT-PCR validation. The primary objective in this specific instance was to perform a technical validation of the overall accuracy and reproducibility of our RNA-seq data. To avoid any potential selection bias during this technical assessment, genes were chosen randomly from the broader pool identified in the sequencing. We fully acknowledge the reviewer's valid point that selecting genes based on their activity levels, biological relevance to the processes under investigation, or integrated analysis of the dataset (as discussed) represents a more biologically informative strategy, particularly for validating specific functional hypotheses. We agree that this targeted approach, leveraging the full dataset synthesis, is significantly superior for future studies focused on mechanistic insights or pathway validation, and we commit to employing this strategy in our subsequent work.

  1. In the Conclusion section, it would be good to add a few suggestions regarding the prospects for further work, exploration of unresolved issues and approaches that can be used for this.

Response: Thanks for your valuable suggestion. We have added one sentence as “This research provides deeper insights into the diversity of ARF genes in ginger and establishes a theoretical framework for future investigations into their functional roles.” in the end of Conclusion.

  1. Lines 188-189 – apparently numbers 293-294 got here by mistake?

Response: Sorry for this mistake. We have corrected this error and revised this sentence as “The abscisic acid response element (ABRE) was present in the promoter regions of all ZoARF genes except ZoARF#14 and ZoARF#19.”

  1. Caption for Figure 6 - if the full Latin name has already been mentioned in the text, an abbreviated name can be used in the captions to the figures.

Response: Thanks for your suggestion. We have amended Figure 6's caption according your suggestion.

  1. In the Discussion and Materials and Methods sections – the same. After the first mention of the full Latin name in the text of the manuscript, the abbreviated one can be used.

Response: Sorry for this mistake. In accordance with your suggestion, we have now systematically implemented abbreviated Latin names throughout the manuscript. Following the first full mention of each species in the main text, abbreviated forms are consistently used in all subsequent references.

  1. Line 339 – typo in the word “reduction”.

Response: Sorry for this mistake.The typographical error ("redunction") identified at Line 339 has been corrected to "reduction" in the revised manuscript. We have verified this correction throughout the text to ensure consistency and accuracy in terminology.

  1. Figure 9b – some unnecessary details are visible in the graph in the lower right corner.

Response: Thanks for your detailed review. The unnecessary details in the lower right corner of the subfigure have been carefully removed in the revised version.

Reviewer 2 Report

Comments and Suggestions for Authors

The article by the authors Yuanyuan Tong, Sujuan Xu, Jiayu Shi, Yi He, Honglei Li, Tian Yu, Sinian Zhang and Haitao Xing “Genome-Wide Analysis and Expression Profiles of Auxin Response Factors in Ginger (Zingiber officinale Roscoe)” is devoted to the study of the family of Auxin Responsive Factors (ARFs) in one of the important economic crops Zingiber officinale.

The article is well written, has scientific significance, the results can be used for a deeper study of the family of Auxin Responsive Factors (ARFs), including their use in breeding.

The article has a number of disadvantages that need to be corrected to prepare the article for publication.

Line 145 in the word "relationship" add the letter "r"

Line 263 Solanum tuberosum write in italics.

In the caption to Figure 2, the authors focus on a color image, so it is necessary to replace the black and white drawing with a color one.

In the text, the authors describe Figure 2 a, b, c, d. Mark parts a, b, c, d in the figure.

Figure 4 consists of three 3 drawings, mark them in the caption, make the corresponding references in the text.

In the description of the results in section 2.10. «Results»: Expression Profiling of Ginger ZoARF Genes in Different Tissues, when setting up the experiment and assessing the level of gene expression, did the authors use a control?

What control did the authors use to obtain the results described in section 2.11. «Results»: Expression Profiles of ZoARF Genes Under Abiotic Stress Conditions?

The text that describes Figure 8 refers to the observation of the time of stress exposure on the expression of ZoARF Genes in different plant tissues.

It is necessary to more clearly indicate the time in the figure and to emphasize in the text the change in gene expression depending on the duration of stress exposure and the effect of plant hormones.

Mark the control in Figure 8.

In the "Materials and Methods" section, the effects of stress and plant hormones on the expression of ZoARF Genes in Zingiber officinale tissues should be described in more detail.

The authors should clearly indicate the selection significance of their work, as they have chosen one of the economically valuable crops, Zingiber officinale, for study. Add this information to the "Introduction", "Conclusion", and "Abstract" sections.

Author Response

Dear reviewer,

We really appreciate the thoughtful review and constructive feedback provided by the reviewers. We agree with the reviewers’ suggestion and have incorporated their suggested changes into the manuscript. In order to increase the readability and quality of the article, the professional editing was performed for grammar and presentation. 

In this revised manuscript, modifications made in response to the reviewer's suggestions are highlighted with a yellow background, while changes addressing errors or inappropriate English grammar and vocabulary are highlighted with a green background.

We make a point-by-point response to reviewers in the bottom of this letter.

We sincerely appreciate the time and effort invested by the reviewers in evaluating our manuscript. We look forward to any additional feedback or suggestions.

Best,

Hai-Tao Xing

xinght@cqwu.edu.cn 

Jul.30, 2025

Reviewer 2

  1. Line 145 in the word "relationship" add the letter "r"

Response: Thank you for your detailed review. We have corrected the typographical error.

  1. In the caption to Figure 2, the authors focus on a color image, so it is necessary to replace the black and white drawing with a color one.

Response: Thank you for pointing this out. The original grayscale image has been replaced with a full-color version in the revised manuscript, ensuring visual alignment with the textual description of color-dependent features.

  1. In the text, the authors describe Figure 2 a, b, c, d. Mark parts a, b, c, d in the figure.

Response: Thank you for pointing this out. We have now clearly marked panels (a), (b), (c), and (d) in the revised figure to ensure precise correspondence between textual descriptions and visual elements. This modification eliminates potential ambiguity and enhances reader navigation through the figure components.

  1. Figure 4 consists of three 3 drawings, mark them in the caption, make the corresponding references in the text.

Response: We appreciate the constructive feedback. In accordance with your suggestion, we have amended Figure 4 to explicitly label its three subfigures as (a), (b), and (c) in the caption. Additionally, we have supplemented the main text with specific references to each subfigure (e.g., "Figure 4(a)", "Figure 4(b)", "Figure 4(c)") at their corresponding discussion points. These revisions ensure clarity in aligning the graphical data with our textual analysis.

  1. In the description of the results in section 2.10. «Results»: Expression Profiling of Ginger ZoARF Genes in Different Tissues, when setting up the experiment and assessing the level of gene expression, did the authors use a control?

Response: Thank you for your comprehensive review. In the expression profiling experiment conducted across twelve ginger tissues, leaf tissue was employed as the reference control, against which all other tissues were compared.

  1. What control did the authors use to obtain the results described in section 2.11. «Results»: Expression Profiles of ZoARF Genes Under Abiotic Stress Conditions?

Response: Thank you for your comprehensive review. In the abiotic stress response assays, the authors utilized the 0-hour sample, collected prior to the application of any treatment, as the control, against which all subsequent time points were compared.

  1. The text that describes Figure 8 refers to the observation of the time of stress exposure on the expression of ZoARF Genes in different plant tissues. It is necessary to more clearly indicate the time in the figure and to emphasize in the text the change in gene expression depending on the duration of stress exposure and the effect of plant hormones.

Response: Thank you for your comprehensive review. Figure 8 illustrates the expression profiles of ARF across 12 distinct tissues, utilizing leaves as the control (CK) for relative differential expression analysis. Figure 9 depicts the differential expression analysis of ARF conducted 12 hours following five abiotic stress treatments, with samples taken from the third leaf from the top and compared to baseline levels (0 h). These data originate from our previous sequencing experiments. Differential expression analysis across different tissues under varying stress conditions was not performed. The timeline mentioned by the reviewer has been further annotated in the figure to improve visual clarity, and the corresponding text descriptions have been revised accordingly.

  1. Mark the control in Figure 8.

Response: Thanks for your valuable suggestion. We have added the CK in Fig.8

  1. In the "Materials and Methods" section, the effects of stress and plant hormones on the expression of ZoARF Genes in Zingiber officinale tissues should be described in more detail.

Response: Thanks for your valuable suggestion. In this study, 4.7 Plant Materials, we provided comprehensive descriptions of the methodologies employed for each stress treatment. The sample of 0-hour (unprocessed) was designated as the control, and subsequent samples were collected 12 hours post-treatment for RNA sequencing (RNA-SEQ) analysis to assess differential gene expression. Due to the absence of significant phenotypic changes within the 12-hour post-treatment period—except in the case of high-temperature stress—the effects of short-term abiotic stress treatments were not evaluated. Nevertheless, both RNA-SEQ and quantitative real-time PCR (qRT-PCR) analyses revealed differential patterns of gene expression.

  1. The authors should clearly indicate the selection significance of their work, as they have chosen one of the economically valuable crops, Zingiber officinale, for study. Add this information to the "Introduction", "Conclusion", and "Abstract" sections.

Response: Thanks for your valuable suggestion. We have now clearly articulated the economic significance of Zingiber officinale (ginger) as the primary justification for its selection. Abstract: We have added the sentence: "Zingiber officinale Roscoe (ginger), an economically important crop, fulfills multifunctional roles as a spice, vegetable, and raw material for medicinal and chemical products." (Lines 11-12). Introduction: The economic significance of ginger as a valuable crop is already described in detail within the existing text (Lines 86-93). Conclusion: We have added the sentence: "Ginger is an important economic crop with both medicinal and edible values." (Line 646). These revisions ensure the economic rationale for selecting ginger is clearly articulated in all key sections of the manuscript.

Reviewer 3 Report

Comments and Suggestions for Authors

Genome-Wide Analysis and Expression Profiles of Auxin Response Factors in Ginger (Zingiber officinale Roscoe)

Yuanyuan Tong, Sujuan Xu, Jiayu Shi, Yi He, Honglei Li, Tian Yu, Sinian Zhang, and Haitao Xing

General observation.

Please verify ZoARF is in italics when referring to the gene. Also, in several parts of the manuscript, there are double spaces between words. In the discussion section, some results are repeated. Would it be possible to merge the results section with the discussion section to avoid repetition? This is just a suggestion. Could you elaborate more on the discussion and the biological significance of the work? There is no discussion about the correlation (or lack thereof) between RNA-seq and qRT-PCR. These changes are substantial and must be made before the article is considered for acceptance.

The manuscript's English requires moderate revision. Please see below for a couple of comments on the abstract suggested by the reviewer.

Specific comments, along with their corresponding line numbers, are listed below.

Line 11: Auxin Response Factors.

Line 35: “the auxin/indole-3-acetic acid (Aux/IAA), Gretchen Hagen 3 (GH3), small auxin-up RNA (SAUR), and auxin response factor (ARF) families”. Gene nomenclature should be in italics; please make the change.

Line 61: “Arabidopsis.” Since it's a genus, it should be written in italics. Please check throughout the manuscript.

Line 124: Where were the sequences for the phylogenetic analysis obtained? This information should be included in the Methods section. A supplementary table listing the accession number or ID of each sequence used, along with the complete sequence, is also required.

Line 125: According to the objective of phylogenetic analysis, what is the significance of finding a distribution of ARFs across four different clades? Is there a specific function of ARFs according to the clade in which they are grouped? Or do they only show relative relationships where they likely perform similar functions across different species?

Line 130: The caption of Figure 1 looks pretty general. The reviewer suggests including information to complete the figure. For example, where the sequences were aligned, where the analysis was performed, and what method was used (bootstrap, etc.).

Line 155: This is confusing; it's impossible to differentiate between UTRs and exons. Authors should verify that the information mentioned in the figure legend matches (note that the figure does not include colors). Check this throughout the entire Figure 2 legend.

Line 208: This is not detailed in the methods section.

Fig. 6: Place scientific names in italics. Why did you choose these models?

Lines 277-280. 283-288: It seems more like a discussion.

Line 280: Since the full name has already been provided, the abbreviation should be used from now on.

Line 296: THE THE repeated.

Line 298: The results of the RNA-seq analysis are missing.

Line 315: Why was the decision made to choose them randomly?

Table S10. Authors should include complete information on the list of primers used. For example, product size and complete sequence, among other factors, are used to ensure reproducibility.

In particular, the reviewer believes that Table S1 should be included in the manuscript, as it is a crucial part of the research, providing relevant information on ARFs in ginger.

Line 487: Did you use a pre-made transcriptome? Where did it come from? The reviewer finds this confusing because, based on the context of the manuscript, the reviewer understands you did the sequencing from scratch. Please explain this.

Line 488: Did you perform differential expression analysis? Nothing is mentioned about methods or results.

Line 524: Please explain better how the seedlings from which the samples were taken were grown and managed. The reviewer also thinks the distribution of treatments is not entirely clear. The reviewer suggests that most details must be provided, or a more precise explanation of what each treatment entails. You could add a supplementary table showing the number of treatments and their content.

Line 526: The reviewer assumes that the expression analysis was done after the bioinformatics analysis, so this part of the methodology should go at the end and not in this line.

Line 530: What was the control treatment?

Line 538: Write in the past.

Line 541: Add the genome citation. Which "previous studies"?

Line 576: Why were these crops chosen?

Line 590: Why did you choose only 12 hours? What samples and how much did you use for RNA-seq? How and where was the sequencing performed? What were the characteristics (platform, depth, mode, etc.)? How was the raw data treated? Was there any cleaning of low-quality sequences? There is no information about the sequencing in the results section. If the transcriptome has not been performed, there is also no information on data handling.

Lines 595, 596, 598: Please add the catalog number.

Line 603: Please add the method citation.

Line 628: Please complete.

Line 643: Place the database where the RNA-seq raw data was deposited.

Abstract. The family of Auxin-Responsive Factors (ARFs) plays a crucial role in facilitating the transduction of auxin signals and is vital to the growth and development of plants. Nevertheless, the role of ARF genes in ginger (Zingiber officinale Roscoe), which holds considerable economic importance, remains to be clarified. In this study, a total of 26 ZoARF genes were identified in the ginger genome, which were further categorized into four subfamilies (I- 15IV) and displayed a non-uniform distribution across 11 chromosomes. The proteins are predominantly localized to the nucleus. The promoter regions harbored numerous response elements associated with light signaling, plant hormones, growth and development, and stress responses. Collinearity analysis revealed nine pairs of fragment duplication events in ZoARFs, all of which were uniformly distributed across their related chromosomes. In addition, the expression profiles of ZoARFs in ginger were analyzed during development and under various stress conditions, including ABA, cold, drought, heat, and salt, using RNA-seq data and quantitative real-time PCR (qRT-PCR) analysis. Notably, expression profiling indicated tissue-specific functions, with ZoARF#04/05/12/22 associated with flower development and ZoARF#06/13/14/23 implicated in root growth. This work provides an in-depth insight into the ARF family and lays the foundation for future investigations into the potential functions of ZoARF genes in ginger growth, development, and abiotic stress tolerance.

Comments on the Quality of English Language

The manuscript's English requires moderate revision. 

Author Response

Dear reviewer,

We really appreciate the thoughtful review and constructive feedback provided by the reviewers. We agree with the reviewers’ suggestion and have incorporated their suggested changes into the manuscript. In order to increase the readability and quality of the article, the professional editing was performed for grammar and presentation. 

In this revised manuscript, modifications made in response to the reviewer's suggestions are highlighted with a yellow background, while changes addressing errors or inappropriate English grammar and vocabulary are highlighted with a green background.

We make a point-by-point response to reviewers in the bottom of this letter.

We sincerely appreciate the time and effort invested by the reviewers in evaluating our manuscript. We look forward to any additional feedback or suggestions.

Best,

Hai-Tao Xing

xinght@cqwu.edu.cn 

Jul.30, 2025

Reviewer 3

The manuscript has been written well but some areas need improvement.

  1. Line 11: Auxin Response Factors.

Response: Sorry for the mistake. We have reviewed the manuscript and corrected these errors in this revision.

  1. Line 35: “the auxin/indole-3-acetic acid (Aux/IAA), Gretchen Hagen 3 (GH3), small auxin-up RNA (SAUR), and auxin response factor (ARF) families”. Gene nomenclature should be in italics; please make the change.

Response: Thanks for the thorough review. We have reviewed the manuscript and corrected these errors in this revision.

  1. Line 61: “Arabidopsis.” Since it's a genus, it should be written in italics. Please check throughout the manuscript.

Response: Thank you for pointing this out. We have carefully checked the manuscript and ensured that all genus names are italicized according to the rule of scientific nomenclature.

  1. Line 124: Where were the sequences for the phylogenetic analysis obtained? This information should be included in the Methods section. A supplementary table listing the accession number or ID of each sequence used, along with the complete sequence, is also required.

Response: Thanks for the thorough review. According to your advice, the Section 4.3 (Lines 569-572) of the Materials and Methods as : "The protein sequences of the ZoARF gene family were analyzed by comparing them with the ARF gene families of Arabidopsis thaliana and Oryza sativa retrieved from the PlantTFDB database (http://planttfdb.gaolab.org/index.php)(Table S1) [68], using MEGA software." This explicitly states that the comparative sequences for Arabidopsis thaliana and Oryza sativa were obtained from the PlantTFDB database. Furthermore, in full accordance with the reviewer's request, a supplementary table listing the accession number/ID and the complete sequence for every gene used in the phylogenetic analysis has been provided as Supplementary Material Table1. This table ensures full transparency and allows for the replication of the analysis.We appreciate the reviewer's attention to detail in ensuring the methods are thoroughly documented.

  1. Line 125: According to the objective of phylogenetic analysis, what is the significance of finding a distribution of ARFs across four different clades? Is there a specific function of ARFs according to the clade in which they are grouped? Or do they only show relative relationships where they likely perform similar functions across different species?

Response: Thanks for the thorough review. Phylogenetic clustering of ARFs into four major clades (I–IV) is highly significant as it reveals evolutionarily conserved functional divergence within this transcription factor family. Clade membership strongly predicts specialized molecular functions, as demonstrated by studies in Arabidopsis thaliana: Clade I (AtARF1/2) regulates flowering time and senescence [1]; Clade II (AtARF3/4) mediates reproductive and vegetative development [2]; Clade III (AtARF10/16/17) governs root development via miRNA160 regulation [3]; and Clade IV (AtARF5-8/19) controls jasmonate signaling and lateral root formation [4]. This clade-specific functional partitioning is largely conserved in ginger, as evidenced by expression patterns. For instance, ginger Clade III genes (ZoARF13/14/23) exhibit root-predominant expression mirroring their Arabidopsis orthologs, while Clade IV members (ZoARF04/05/12/22) align with floral functions. Notably, exceptions like ZoARF06 (Clade II) show divergent root-specific expression in ginger—unlike its Arabidopsis counterparts—suggesting potential neo-functionalization adapted to rhizome-based growth. Thus, phylogenetic analysis provides a robust framework for predicting gene functions across species while highlighting lineage-specific innovations.

  • Ellis, C.M.; Nagpal, P.; Young, J.C.; Hagen, G.; Guilfoyle, T.J.; Reed, J.W. AUXIN RESPONSE FACTOR1 and AUXIN RESPONSE FACTOR2 regulate senescence and floral organ abscission in Arabidopsis thaliana. Development. 2005, 132, 4563-4574. https://doi:10.1242/dev.02012.
  • Pekker, I.; Alvarez, J.P.; Eshed, Y. Auxin Response Factors Mediate Arabidopsis Organ Asymmetry via Modulation of KANADI Activity. The Plant Cell. 2005, 17, 2899–2910. https://doi.org/10.1105/tpc.105.034876.
  • Tabata, R.; Ikezaki, M.; Fujibe, T. Arabidopsis auxin response factor6 and 8 regulate jasmonic acid biosynthesis and floral organ development via repression of class 1 KNOX genes. Plant Cell Physiol. 2010, 51, 164-175. https://doi:10.1093/pcp/pcp176
  • Okushima, Y.; Overvoorde, P.J.; Arima, K. Functional genomic analysis of the AUXIN RESPONSE FACTOR gene family members in Arabidopsis thaliana: unique and overlapping functions of ARF7 and ARF19. Plant Cell. 2005, 17, 444-463. https://doi:10.1105/tpc.104.028316
  1. Line 130: The caption of Figure 1 looks pretty general. The reviewer suggests including information to complete the figure. For example, where the sequences were aligned, where the analysis was performed, and what method was used (bootstrap, etc.).

Response: We sincerely thank the reviewer for the constructive suggestion. As recommended, we have revised the caption of Figure 1 to include essential methodological details for better clarity and reproducibility. We rewrote it as ” Phylogenetic analysis of ARF protein in A. thaliana, O. sativa, and Z. officinale. This analysis employed ARF protein amino acid sequences from these species and was constructed using the maximum likelihood (ML) method in MEGA11. The numbers at the branches indicate the confidence values obtained from the 1000 bootstrap tests. Roman numerals I–IV represent different ARF groups.”

  1. Line 155: This is confusing; it's impossible to differentiate between UTRs and exons. Authors should verify that the information mentioned in the figure legend matches (note that the figure does not include colors). Check this throughout the entire Figure 2 legend.

Response: We sincerely thank the reviewer for their valuable observation regarding the clarity and consistency of Figure 2 and its legend. We have carefully revised Figure 2 to address these concerns: Color Implementation: The figure itself has now been generated with the specific colors described in the legend (green for UTRs, yellow for exons). Enhanced Clarity: To further aid differentiation between UTRs and exons, direct labels have been added to the relevant parts of the gene structure diagram (Figure 2b). The attached revised Figure 2 and its corresponding legend reflect these essential corrections. We appreciate the reviewer's diligence in ensuring the accuracy and clarity of our presentation.

  1. Line 208: This is not detailed in the methods section.

Response: Thank you for your valuable advice. This part we have rewrote as “The cis-acting elements within the promoter regions of ginger ZoARF genes were investigated by analyzing the 2,000 bp upstream sequences of ZoARFs using the Plant-CARE online database [72]. Subsequently, the cis-acting elements were categorized and tallied, and presented as histograms using SigmaPlot software. Furthermore, heatmaps depicting distinct categories of cis-acting elements were constructed using TBtools.” (Lines 582-587).

  1. 6: Place scientific names in italics. Why did you choose these models?

Response: Thanks for the thorough review. We have ensured that all scientific names (Musa acuminata, Oryza sativa, Arabidopsis thaliana, Solanum tuberosum, and Zingiber officinale) in Figure 6 are now italicized as per standard convention. The four comparative species were strategically selected to enable robust evolutionary analysis of ginger ARF genes: Musa acuminata (banana) as the closest monocot relative to ginger (Zingiber officinale) for assessing conserved synteny; Oryza sativa (rice) as a divergent monocot model for broader phylogenetic comparison; Arabidopsis thaliana as a universal reference dicot; and Solanum tuberosum (potato) as a functionally relevant dicot with storage tubers analogous to ginger rhizomes. This comparative framework successfully highlighted the divergent evolution of monocot and dicot ARF lineages, underscored by the high number of orthologous ARF pairs identified between ginger and banana (36 pairs) in contrast to the absence of collinearity observed with the dicot species.

  1. Lines 277-280. 283-288: It seems more like a discussion.

Response: We sincerely appreciate the reviewer's valuable feedback regarding the placement of the results describing the PPI network interactions (Lines 277-280, 283-288). We respectfully maintain that these descriptions are most appropriate within the Results section. This text primarily reports the direct topological findings from the STRING network analysis, specifically detailing the observed and predicted interactions between the ZoARF proteins and key partners (AUX1, BZR1, PIF4, CRL1), including connectivity metrics like AUX1's interaction with six ZoARFs. The brief functional context cited from references [35-37] for BZR1, PIF4, and CRL1 is included solely to provide essential biological justification for why these specific interaction partners are significant within the scope of our results. Therefore, we believe this content accurately presents the core network analysis outcomes and have retained it in the Results section. We thank the reviewer for raising this important point for consideration.

  1. Line 280: Since the full name has already been provided, the abbreviationshould be used from now on.

Response: Thank you for pointing this out. We have reviewed the manuscript to ensure the abbreviation of related protein were used from this line.

  1. Line 296: THE THE repeated.

Response: Thank you for your detailed review and for catching this typographical error. The repeated "THE" was an inadvertent oversight during editing and has now been corrected to a single "THE" in the revised manuscript. We appreciate your careful reading and attention to detail, which has strengthened the clarity of the text. We have also performed an additional proofread of the entire manuscript to ensure no similar errors remain.

  1. Line 298: The results of the RNA-seq analysis are missing.

Response: We appreciate the reviewer's attention to detail regarding the RNA-seq analysis. In this study, the expression profiles of ARF genes were determined using existing, relevant sequencing datasets from other projects, which provided robust and sufficient data for our specific gene screening objectives. We rewrite the Data Availability Statement(Line 686-687 ) as“The resulting transcriptome data have been archived in the NCBI Sequence Read Archive under accession number SRP476742.”

  1. Line 315: Why was the decision made to choose them randomly?

Response: Thanks for the thorough review. To validate our RNA-seq data, we randomly chose genes from the tissue expression and stress sections for qRT-PCR analysis. After 12 hours of treatment, the qRT-PCR results aligned with the RNA-seq findings, showing similar trends in gene expression changes.

  1. Table S10. Authors should include complete information on the list of primers used. For example, product size and complete sequence, among other factors, are used to ensure reproducibility.

Response: Thank you for your valuable suggestion. We sincerely appreciate the reviewer's valuable suggestion regarding the inclusion of comprehensive primer details in Table S10. We have supplemented Table S10 with the complete primer sequences and product sizes, as suggested.

  1. In particular, the reviewer believes that Table S1 should be included in the manuscript, as it is a crucial part of the research, providing relevant information on ARFs in ginger.

Response: We thank the reviewer for highlighting the importance of Table S1 (detailing the relevant ARF information in ginger) and agree that its inclusion within the main manuscript significantly enhances the accessibility and context of this crucial data. Accordingly, we have incorporated this table into the main body of the manuscript as Table 1.

  1. Line 487: Did you use a pre-made transcriptome? Where did it come from? The reviewer finds this confusing because, based on the context of the manuscript, the reviewer understands you did the sequencing from scratch. Please explain this.

Response: We appreciate the reviewer's careful attention to this detail and apologize for any confusion regarding the transcriptome data source. To clarify: we utilized a pre-existing transcriptome dataset (NCBI SRA accession: SRP476742) , not newly generated sequencing data, for the analysis referenced at Line 487. We have now explicitly stated this in the revised manuscript and included the dataset accession in the Data Availability Statement to ensure transparency.

  1. Line 488: Did you perform differential expression analysis? Nothing is mentioned about methods or results.

Response: Thank you for your query regarding the differential expression analysis (DEA). We confirm that standard DEA was indeed performed on the transcriptome data as part of our comprehensive bioinformatic processing. However, as the primary focus of our study was the specific expression patterns and functional characterization of ARF genes, detailed methodological descriptions and results of the broader transcriptome-wide DEA fall outside the central scope of this manuscript. Consequently, we prioritized the presentation and discussion of results directly pertaining to the ARF family. We appreciate your attention to methodological detail and are happy to provide supplementary information on the full DEA pipeline upon request.

  1. Line 524: Please explain better how the seedlings from which the samples were taken were grown and managed. The reviewer also thinks the distribution of treatments is not entirely clear. The reviewer suggests that most details must be provided, or a more precise explanation of what each treatment entails. You could add a supplementary table showing the number of treatments and their content.

Response: Thanks for the thorough review. We rewrote this section as “The study examined the ginger cultivar "Laiwu No. 2," provided by Shandong Second Academy of Agricultural Sciences, China. The seed ginger, characterized by healthy buds, was divided into approximately 30-gram pieces, each with 2-3 buds, and planted in 40 cm by 20 cm pots filled with a 6:3:1 mix of peat, garden soil, and perlite. The seedling are grown in a greenhouse with 16/8 hours of light/darkness daily. The expression of the ZoARF genes was examined in seedlings approximately 6 months old, encompassing various plant parts such as leaves (particularly the third spot from the root tip towards the base of the stem), roots, leaf buds, rhizome buds, flower buds, mature flowers, the base of the stem, flower stalk, and the rhizome internode -1st, -2nd, and -3rd. Two-month-old seedlings were subjected to salt and drought stress using 200 mM NaCl and 15% PEG6000 solutions to study the gene's response to abiotic stress. Additionally, a 0.1 mM ABA solution was applied to the ginger leaves. At 40 ℃ and 4 ℃, respectively, the heat and cold stressors were addressed. After subjecting the plants to salt, low temperature, and drought treatments, leaf samples were collected at different intervals: 0, 1, 3, 6, 12, 24 and 48 hours. Each response was given three times. At specific time intervals of 0, 1, 3, 6, 12, and 24 hours, leaf samples were collected for the purpose of heat treatment (Table S10). The collected samples were immediately frozen in liquid nitrogen and stored at -80 °C.” in Line 609-626.

  1. Line 526: The reviewer assumes that the expression analysis was done after the bioinformatics analysis, so this part of the methodology should go at the end and not in this line.

Response: We appreciate the reviewer's insightful observation regarding the placement of the expression analysis methodology. The reviewer correctly noted that this analysis logically follows the bioinformatics analysis. Accordingly, we have relocated the description of the expression analysis to the end of the Methods section, as suggested in line 609.

  1. Line 530: What was the control treatment?

Response: Thank you for pointing this out. In our experimental design, the control (CK) consisted of seedlings that were left completely untreated and harvested immediately at 0 h. All stress treatments (salt, drought simulation, ABA, low temperature and high temperature) were imposed on comparable seedlings that had been grown under identical conditions up to that same 0-h time point. Consequently, the CK represents the true baseline physiological and molecular status of the plants before any external factor was applied.

  1. Line 538: Write in the past.

Response: Thank you for highlighting this point. We acknowledge that the description of sample handling in Line 538 ("The gathered samples are kept at -80 °C and are instantly frozen in liquid nitrogen") inadvertently used the present tense. This has been corrected to consistently use the past tense throughout the Methods section to accurately reflect completed experimental procedures. The revised sentence now reads: "The collected samples were immediately frozen in liquid nitrogen and stored at -80 °C." We appreciate your careful attention to detail in ensuring grammatical consistency.

  1. Line 541: Add the genome citation. Which "previous studies"?

Response: We appreciate the reviewer’s attention to precision. Upon careful reevaluation, we agree that the statement "previous studies have revealed that the ginger genome contains a significant diversity of ARF genes" does not directly contribute to the original findings presented in this work. To maintain focus on our genomic analysis and avoid ambiguity regarding external sources, we have removed this sentence entirely in the revised manuscript. This modification enhances the precision of our narrative and ensures all claims are fully supported by data generated in this study.

  1. Line 576: Why were these crops chosen?

Response: Thank you for your feedback. The rationale for selecting these crops has been comprehensively addressed in our response to Question 9 within the revised manuscript.

  1. Line 590: Why did you choose only 12 hours? What samples and how much did you use for RNA-seq? How and where was the sequencing performed? What were the characteristics (platform, depth, mode, etc.)? How was the raw data treated? Was there any cleaning of low-quality sequences? There is no information about the sequencing in the results section. If the transcriptome has not been performed, there is also no information on data handling.

Response: In reference to existing literature, several studies have employed a 12-hour stress treatment under adverse conditions followed by RNA-SEQ analysis. Our decision to utilize a 12-hour sampling period was informed by preliminary experiments, which revealed the following: (1) During a 48-hour high-temperature exposure at 40°C, nearly all leaves desiccated, with 30%-40% of leaves exhibiting dryness at 24 hours, whereas only a minor fraction (less than 5%) showed signs of dryness at 12 hours. No substantial leaf damage was observed at 6, 3, and 1-hour intervals, thus ensuring leaf integrity and identifying 0, 1, 3, 6, and 12 hours as optimal sampling intervals. (2) Initial laboratory investigations indicated that stress response genes exhibited significant differential expression predominantly between 8 and 12 hours. (3) Given that sample collection requires 1-2 minutes, sampling at the 1-hour mark could introduce considerable variability between samples collected at slightly different times. Consequently, based on these considerations, we selected the 12-hour post-stress treatment interval for sequencing analysis. Alternatively, this approach aligns with methodologies employed in prior studies, as documented in the literature. 

The sequencing data pertaining to our analyses of ginger tissue expression and stress treatments have been previously measured, analyzed, and archived in the NCBI database. As our objective was to utilize pre-existing transcriptome data instead of undertaking a specialized transcriptome analysis, we initially provided a description of the treatment process without elaborating on the specific sequencing methodologies or platform specifications. If the sequencing method and platform need to be described, we weill supplement this information in the revised manuscript.

  1. Lines 595, 596, 598: Please add the catalog number.

Response: Thanks for pointing this out. We have added the catalog number in line 633-637.

  1. Line 603: Please add the method citation.

Response: We thank the reviewer for highlighting the need for methodological citation. As suggested, we have supplemented the method description at line 643 with an appropriate reference to support the experimental approach. This addition ensures the methodological rigor is properly contextualized within existing literature. We believe this revision adequately addresses the comment.

  1. Line 628: Please complete.

Response: Thank your for point this out. The online version includes the supplementary material. During the review stage, we were unable to complete this sentence accurately, so we have now removed it.

  1. Line 643: Place the database where the RNA-seq raw data was deposited.

Response: Thank you for pointing this out. We add one sentence as “The resulting transcriptome data have been archived in the NCBI Sequence Read Archive under accession number SRP476742.” in line 686.

Round 2

Reviewer 3 Report

Comments and Suggestions for Authors

The manuscript was substantially improved, both in English and content.